# Comparison of the Behavior of Perivascular Cells (Pericytes and CD34+ Stromal Cell/Telocytes) in Sprouting and Intussusceptive Angiogenesis

**DOI:** 10.3390/ijms23169010

**Published:** 2022-08-12

**Authors:** Lucio Díaz-Flores, Ricardo Gutiérrez, Maria Pino García, Miriam González-Gómez, Lucio Díaz-Flores, Jose Luis Carrasco, Juan Francisco Madrid, Aixa Rodríguez Bello

**Affiliations:** 1Department of Basic Medical Sciences, Faculty of Medicine, University of La Laguna, 38071 Tenerife, Spain; 2Department of Pathology, Eurofins Megalab–Hospiten Hospitals, 38100 Tenerife, Spain; 3Instituto de Tecnologías Biomédicas de Canarias, University of La Laguna, 38071 Tenerife, Spain; 4Department of Cell Biology and Histology, School of Medicine, Campus of International Excellence “Campus Mare Nostrum”, IMIB-Arrixaca, University of Murcia, 30120 Murcia, Spain; 5Department of Bioquímica, Microbiología, Biología Celular y Genética, University of La Laguna, 38071 Tenerife, Spain

**Keywords:** angiogenesis, pericytes, telocytes, stromal cells, endothelial cells, sprouting angiogenesis, intussusceptive angiogenesis

## Abstract

Perivascular cells in the pericytic microvasculature, pericytes and CD34+ stromal cells/telocytes (CD34+SCs/TCs), have an important role in angiogenesis. We compare the behavior of these cells depending on whether the growth of endothelial cells (ECs) from the pre-existing microvasculature is toward the interstitium with vascular bud and neovessel formation (sprouting angiogenesis) or toward the vascular lumen with intravascular pillar development and vessel division (intussusceptive angiogenesis). Detachment from the vascular wall, mobilization, proliferation, recruitment, and differentiation of pericytes and CD34+SCs/TCs, as well as associated changes in vessel permeability and functionality, and modifications of the extracellular matrix are more intense, longer lasting over time, and with a greater energy cost in sprouting angiogenesis than in intussusceptive angiogenesis, in which some of the aforementioned events do not occur or are compensated for by others (e.g., sparse EC and pericyte proliferation by cell elongation and thinning). The governing mechanisms involve cell–cell contacts (e.g., peg-and-socket junctions between pericytes and ECs), multiple autocrine and paracrine signaling molecules and pathways (e.g., vascular endothelial growth factor, platelet-derived growth factor, angiopoietins, transforming growth factor B, ephrins, semaphorins, and metalloproteinases), and other factors (e.g., hypoxia, vascular patency, and blood flow). Pericytes participate in vessel development, stabilization, maturation and regression in sprouting angiogenesis, and in interstitial tissue structure formation of the pillar core in intussusceptive angiogenesis. In sprouting angiogenesis, proliferating perivascular CD34+SCs/TCs are an important source of stromal cells during repair through granulation tissue formation and of cancer-associated fibroblasts (CAFs) in tumors. Conversely, CD34+SCs/TCs have less participation as precursor cells in intussusceptive angiogenesis. The dysfunction of these mechanisms is involved in several diseases, including neoplasms, with therapeutic implications.

## 1. Introduction

Pericytes and CD34+ stromal cells/telocytes (CD34+SCs/TCs) participate in angiogenesis. Pericytes are flattened or stellate-shaped perivascular cells located in the pericytic microvasculature (precapillary arterioles, capillaries, post-capillary venules, and some venules). They are extensively branched cells that incompletely envelop the endothelium and are embedded within the microvascular basement membrane, except at points in direct contact with endothelial cells (ECs). Pericytes have progenitor capacity and participate in vascular tone, transport and permeability regulation, immunologic defense, coagulation, extracellular matrix formation, cell interaction, angiogenesis, and vessel stabilization [1,2,3].

CD34+SCs/TCs are located in the stroma of multiple anatomical sites, including the perivascular region of the pericytic microvasculature and the adventitia of larger vessels. These cells show a small somatic body and long, moniliform cytoplasmic processes (telopodes) with alternation of slender segments (podomers) and dilations (podoms) [4,5]. In addition to intercellular communication, several roles have been hypothesized for CD34+SCs/TCs, including control and organization of the extracellular matrix, structural support, endocytosis, creation of tissular microenvironments, guidance to cell migration, contribution of scaffolds, immunomodulation, neurotransmission, control and regulation of other cell types, stem cell modulation, and participation in angiogenesis, regeneration, repair, and tumor stroma formation [4,5,6,7,8,9,10,11,12,13,14,15,16,17,18,19,20,21].

Sprouting and intussusceptive angiogenesis are the two principal multi-step processes by which blood vessels form anew from a pre-existing vasculature [22,23]. The mechanisms of neovessel formation are different in these processes. Thus, the neovessels are originated (a) by vessel sprouts that grow outward in sprouting angiogenesis [24] and (b) through the formation of intravascular/transluminal tissue pillars that split and remodel the vessels in intussusceptive angiogenesis [22,25,26]. In sprouting angiogenesis, the overlapping sequential findings include migration of ECs, changes in extracellular matrix with basement membrane degradation, proliferation of ECs, mobilization and proliferation of pericytes and perivascular CD34+SCs/TCs, participation of inflammatory cells, tubulogenesis (vascular lumen development), recruitment of pericytes, formation of a new basement membrane and vessel fusion, pruning, and stabilization [27,28,29]. The main mechanism in intussusceptive angiogenesis is the formation of intravascular tissue folds and pillars (columns or posts), which partially or totally divide the vessel lumen and increase or remodel the vascular network [22,26,30,31,32,33,34,35,36,37]. Microvascular growth, vessel arborization, branching remodeling, and vessel segmentation from a pre-existing microvasculature can occur in intussusceptive angiogenesis [25,26,38,39,40,41,42,43,44,45,46].

Therefore, pericytes and CD34+SCs/TCs are the perivascular cells in the pericytic vasculature and participate in angiogenesis. There are numerous studies on the role of pericytes and CD34+SCs/TCs in sprouting angiogenesis, including reviews of pericytes in general, in which their role in this regard is also outlined [1,47,48], as well as contributions on CD34+SCs/TCs in this field [49,50,51,52,53,54,55,56,57,58,59,60,61]. Less attention has been paid to these cells in intussusceptive angiogenesis and to comparing their behavior according to both types of angiogenesis.

Given the above, the objective of this review is to compare the behavior of pericytes and CD34+SCs/TCs depending on whether angiogenesis occurs by sprouting or intussusception.

## 2. Identification of Pericytes and CD34+SCs/TCs in the Microvasculature

There are no specific or generally stable markers for pericytes and CD34+SCs/TCs [1,48,53,62,63]. Pericytes and the surrounding CD34+SCs/TCs in the pericytic microvasculature form two layers, which can, respectively, be considered as a continuation of the medial layer made up of vascular smooth muscle cells and the external layer or adventitia made up of CD34+SCs/TCs in vessels of greater caliber.

Therefore, cells in a stabilized pericytic microvasculature can be identified by their layer location, above all in semithin and ultrathin sections, in which ECs are present in the intima, SCs/TCs in the external layer, and pericytes are sandwiched between both (Figure 1A). In addition, pericytes and ECs share a basement membrane, whereas SCs/TCs have none (Figure 1A). Although there is no specific marker for pericytes and CD34+SCs/TCs, a relatively wide immunophenotypic profile allows for their identification and distinction from ECs. Thus, CD34+SCs/TCs are CD34+, PDGFRα+, vimentin+, and CD31−; ECs are CD34+, PDGFRA−, vimentin+, and CD31+ [64]; and pericytes, depending on the vascular beds, are αSMA+, NG2+, PDGFRB+, vimentin+, CD13+, CD146+, endoglin (CD105), aminopeptidase A+, and aminopeptidase N+ [65,66,67,68,69,70,71]. In addition, pericytes and CD34+SCs/TCs lose some of their markers when isolated and cultured on plastic surfaces [66].

The combined location of these cells and use of CD34, CD31, and αSMA can meet the requirements to identify the three components of the stabilized microvasculature, since CD34 and CD31 are co-expressed in ECs but not in SCs (CD34+SCs/TCs do not express CD31) [72], and αSMA is expressed in pericytes. In addition, ECs (above all their luminal surface) express CD34 marker more intensely than CD34+SCs/TCs.

Likewise, since CD31 negativity rules out that putative CD34+SCs/TCs are not ECs, the use of double staining with CD34 and SMA is an especially useful and very affordable procedure for following the perivascular cell changes associated with angiogenesis, as well as the presence or not of myofibroblasts and their perivascular source during repair, fibrosis, and tumor stroma formation. This usefulness is increased in processes in which there is intensive sprouting angiogenesis (e.g., during the first stages of granulation tissue formation) (Figure 1B), or strong intussusceptive angiogenesis (e.g., in some reactive vascular processes, such as intravascular papillary endothelial hyperplasia) [73].

## 3. Pericytes and CD34+SCs/TCs in Sprouting Angiogenesis

In sprouting angiogenesis, pericytes and CD34+SCs/TCs show intense morphologic and functional changes, as well as modifications in their arrangement. Next, we consider the behavior of pericytes, followed by that of CD34+SCs/TCs.

## 4. Pericytes in Sprouting Angiogenesis

During sprouting angiogenesis, pericytes (a) establish interactions with ECs, stromal cells, inflammatory cells, and extracellular matrix, (b) detach from the vessel wall, mobilize, proliferate, and are newly recruited, and (c) participate in vessel development, stabilization, maturation, and regression [68,74,75,76,77,78,79,80,81,82,83,84,85,86]. Pericyte-fibroblast transition has also been described [87].

In very early stages of sprouting angiogenesis, pericytes in the parent vessels increase their somatic volume, shorten their processes, and show voluminous nuclei and prominent nucleoli (Figure 2A–C), as well as numerous intracytoplasmic ribosomes, either singly or in aggregates (Figure 2B) [75]. VEGF facilitates the loss of pericyte coverage [88] and induces overexpression of Angiopoietin-2 in ECs, leading to vessel destabilization [89,90] and EC migration [91,92]. Mitoses are observed in pericytes, which show a high proliferative index (Figure 2D,E). EC sprouts have been described with or without loss of pericytes. In the first case (Figure 2F), there is a “plasticity window” [93], with pericyte/EC basement membrane degradation, pericyte detachment from the vessel wall (see above), endothelial tip cell selection and lateral inhibition, and EC migration. Thus, endothelial tip cells [94], lider migrating (invading) cells, are selected (only ECs exposed to the highest VEGF levels become tip cells), degrade cell–cell junctions, and contribute to extracellular matrix proteolysis, in which extracellular-matrix-degrading podosomes on EC surface [95] and metalloproteinases participate [96,97,98,99,100,101]. Endothelial tip cell filopodia, which sense attractant (mainly VEGF-A) and repellent (mainly semaphorin) signals, guide polarized migration [94,102,103,104,105,106,107]. Lateral inhibition promotes stalk cell characteristics in immediately neighboring ECs. In addition, crosstalk between ECs and pericytes in the neovessels directs the site-specific expression of metalloproteinases MT1-MMP to endothelial tip cells [108]. In the second case, when there is no loss of pericytes, they extend beyond sprouting ECs and guide EC migration, determining the location of the newly formed vessels [68,109,110,111]. Pericytes, and occasionally macrophages, can form tubes, which behave as scaffolds for later penetration of ECs [112,113,114,115,116].

During the sprouting phase, pericytes are recruited in the vascular sprouts, and pericyte investment generally occurs during tube formation. These findings are very evident in semithin and ultrathin sections (Figure 3). Several pathways act in the mechanism of pericyte recruitment and vessel stabilization. PDGF-beta plays an important role. This soluble factor is released by ECs and activates the tyrosine Kinase PDGFR beta in pericytes [117]. Other positive or negative regulators can also participate in this process, including TGF-beta, angiopoietins-1 and -2, sphingosine-1, nitric oxide, semaphorin-3A, HB-EGF, and matrix metalloproteinases [118,119,120,121,122,123,124,125,126,127]. Pericyte coverage and above all pericyte–EC maturation occurs during pericyte recruitment [122], leading to EC survival, capillary assembly and basement membrane deposition, vessel stabilization, diameter regulation, and vascular remodeling [77,125,127,128,129,130]. Thus, EC proliferation is regulated by pericytes [131,132,133]. Pericyte contractility is the most important factor that controls EC cycle progression and sprouting [132]. In addition, recruited pericytes also act by stimulating EC maturation by releasing paracrine factors, including angiopoietin 1 and TGF-beta [134].

The newly formed vessels regress in varying numbers, depending on the type of process. Among other factors that participate in vascular pruning and regression are vascular flow, oxygenation, WNT and Notch signaling, and ANG/TIE signaling [135]. The role of pericytes on vessel regression is controversial: mainly whether they undergo apoptosis, pass to other persistent vessels, or remain in situ, and whether vessel regression is dependent on or independent of pericytes [93,136,137,138,139,140,141]. We have observed persistent pericytes after vessel regression, as well as vessels in regression with apparent apoptosis of ECs and pericytes.

Pericytes can be involved in microvascular dysfunction in several pathologic processes, such as infarction, hypertension, diabetes mellitus, sepsis, and neoplasms. For example, in tumor angiogenesis, pericytes show modified morphology, with aberrant processes, may be loosely attached, with different coverage, as well as present changes in their markers [68,142,143,144,145,146]. Thus, when there is low pericyte coverage, tumor cell invasion/extravasation and metastasis is facilitated, although vessel integrity and tumor growth are disturbed [147,148]. Conversely, high pericyte coverage has been observed in some tumors, such as renal cell carcinoma and glioblastoma, with intense pericyte proliferation in the latter [149].

## 5. CD34+SCs/TCs in Sprouting Angiogenesis

During sprouting angiogenesis, perivascular CD34+SCs/TCs dissociate from their perivascular niche, are arranged between a provisional matrix with fibrin and fibronectin, proliferate, acquire transitional cell forms, and differentiate [11,62,150,151,152].

Although CD34+SCs/TCs initially retain a perivascular location in the parent vessels, these cells are larger, and their increased somatic body, with a plump, stellate, or fusiform morphology, is usually more distant from the vascular wall (Figure 4A–D). However, in small vessels, hypertrophied perivascular CD34+SCs/TCs can remain in their original location (Figure 4E). Thus, CD34+SCs/TCs, with long processes, move away from their perivascular niche and tend to surround perivascular edematous spaces in which inflammatory cells are also observed. The nuclei of CD34+SCs/TCs are voluminous, showing one or two prominent nucleoli. Mitoses are seen in CD34+SCs/TCs (Figure 5A,C), and a high proliferative index is observed (Figure 5B–D). Ultrastructurally, the organelles of synthesis, endoplasmic reticulum, and Golgi complex are increased in the cytoplasm of stromal cells, even when they are in mitosis (Figure 5F,G). Extracellular vesicles, exosomes, ectosomes, and multivesicular bodies from CD34+SCs have been observed [153,154], and secretion of VEGF, bFGF, PDGF-alfa, and IL-8 has been demonstrated in these cells, promoting angiogenesis [155,156]. An important example of CD34+SCs/TCs are adipose-derived stromal cells (ASCs), which have been extensively studied. ASCs induce EC migration and proliferation by paracrine secretion, with angiogenic effects [157,158,159,160]. In addition, they facilitate tube and capillary network formation [161,162,163,164]. The ASC paracrine secretion involves VEGF, FGF-2, PDGF, TGFB, and HGF [165,166,167].

In the initial phase of sprouting angiogenesis, perivascular CD34+SCs/TCs maintain the expression of CD34, while the perivascular cells that show positivity for αSMA are pericytes and vascular smooth muscle cells (Figure 4). Some of these SCs present intracytoplasmic lipid droplets, acquiring characteristics reminiscent of alveolar lipofibroblasts, lipid-laden cells, or lipid-interstitial cells. The lipid droplets can be demonstrated in semithin (Figure 6A–D) and ultrathin sections (Figure 6E), and the lipofibroblast-like cells and their progenitors have a proliferative capacity (Figure 6D), express PDGFR alpha (as occurs with CD34+SCs/TCs), and can differentiate into myofibroblasts (Figure 6E) [168,169].

CD34 expression is progressively lost in CD34+SCs/TCs, with successive CD34+SC/TC mitoses. Therefore, a progressive decrease in the intensity of the CD34 labeling of these cells is observed as sprouting angiogenesis advances. As the loss of CD34 positivity is accentuated in CD34+SCs/TCs, gain of αSMA occurs in these SCs, which acquire characteristics of contractile myofibroblasts (Figure 7). Thus, CD34+SCs/TCs and αSMA+SCs, with a varying intensity of expression of these markers, coincide in relatively early and intermediate phases of the evolution of sprouting angiogenesis (Figure 7A). In more advanced stages, all the SCs may correspond to myofibroblasts (Figure 7B). These changes occur mostly in the periphery of the cytoplasm, where these markers are predominantly expressed. Thus, peripheral expression of CD34 in CD34+SCs/TCs is replaced by peripheral αSMA expression in myofibroblasts as bands parallel to the cell’s longitudinal surface, which ultrastructurally correspond to peripheral bands of filaments with dense bodies. Immunochemistry observation suggests the transition between CD34+SCs/TCs and myofibroblasts (Figure 7C,D), which is demonstrated by immunofluorescence labeling. Thus, immunofluorescence labeling of CD34 and αSMA in confocal microscopy has revealed co-location of both markers in some SCs in this type of angiogenesis during granulation tissue and tumor stroma formation (Figure 7E–G) [11,62,150,151,152]. Likewise, as mentioned above, lipofibroblasts originating from CD34+SCs/TCs can differentiate into myofibroblasts, as occurs with alveolar lipofibroblasts in lung fibrosis [168,169].

These findings reveal that resident perivascular CD34+SCs/TCs have progenitor capacity, an ability that can be developed during sprouting angiogenesis. Indeed, in multiple processes, including repair, tumor stroma formation, diabetes, fibrosis and systemic sclerosis, the presence or absence of CD34+SCs/TCs plays an important role. Examples include some malignant neoplasms and systemic sclerosis, in which derangement of the microvascular system occurs [170,171,172,173,174,175,176,177]. In addition to lipofibroblasts and myofibroblasts, CD34+SCs/TCs can be a source of lipoblasts, chondroblasts, and osteoblasts.

Regression of the newly formed vessels can be very high during the rising number of stromal cells in the granulation tissue. Many of the regressing vessels present marked intravascular accumulation of platelets (Figure 8). Thus, the aggregated factor-releasing platelets facilitate stromal cell growth in the granulation tissue, which behaves in this phase as a “paracrine transitional organ” [178].

## 6. Pericytes and CD34+SCs/TCs in Intussusceptive Angiogenesis

The main mechanism of intussusceptive angiogenesis leading to vessel division is the formation of intravascular pillars (inward growth), unlike sprouting angiogenesis, in which vessel sprouts grow toward the interstitium (outward growth). To exclude structures that can simulate intravascular pillars, a 3D demonstration procedure is required, such as vascular corrosion casting using scanning electron microscopy, serial semithin and/or ultrathin sections, intravascular injection of fluorescent dyes, and in vivo microscopic video analysis, or immunofluorescence labeling for endothelial markers in tissue sections using confocal laser scanning microscopy (Figure 9A–D). Intravascular pillars are formed by a cover and a core. The pillar cover is made up of a layer of ECs, and the pillar core is formed by interstitial tissue structures, which may include pericytes (Figure 9E) and other interstitial cells, depending on pillar diameter and the evolutive stage of pillar formation. The trajectory of the pillars is usually incompletely seen in 2D observations, presenting longitudinal, oblique, or transversely sectioned areas. Double immunochemistry or immunofluorescence labeling for CD34 and αSMA, as well as observation in semithin and ultrathin sections, is useful for studying pillar components (Figure 9E, Figure 10 and Figure 11). Given the different type of growth—outward or inward from the parent vessel—in sprouting and intussusceptive angiogenesis, the response in the interstitium and the behavior of pericytes and CD34+SCs/TCs is also different in both types of angiogenesis. For example, interstitial tissue degradation, tissue repair, and granulation tissue formation are minimal in intussusceptive angiogenesis.

The main factors that influence intussusceptive angiogenesis include several molecules, hypoxia, and hemodynamic changes, above all shear stress [25,38,179,180,181,182,183,184]. The molecules involved in this process include VEGF, PDGF B, Notch signaling, endoglin/CD105, EphrinB2/EphrinB4, nitric oxide, and EC MMT1-MMP [185,186,187,188,189,190,191,192]. VEGF may act at low levels, which explains the persistence of angiogenesis after anti-VEGF therapy [193,194,195], or by very high expression, blocking the formation of the gradient for endothelial tip cell migration [188,189,190,191]. PDGF B accelerates splitting angiogenesis, limits pericyte loss induced by high levels of VEGF, and modulates VEGFR2 [196,197,198]. Inhibition of endoglin/CD10 [190] and EphrinB2/EphrinB4 signaling induce [190] and modulate [189] intussusception, respectively. Finally, nitric oxide inhibition facilitates the transition of sprouting angiogenesis to intussusceptive angiogenesis, and EC MMT1-MMP induces nitric oxide production [199].

Next, we consider pericyte and CD34+SC/TC behavior in intussusceptive angiogenesis.

## 7. Pericytes in Intussusceptive Angiogenesis

Pericytes participate in intravascular pillar formation, contributing to vascular division by intussusception. The timing of pericyte incorporation into the pillar depends on the mechanism of pillar formation by (a) the establishment of endothelial contacts between opposite vessel walls and interendothelial bridge development, (b) pillar splitting, (c) the merging of adjacent capillaries and modifications of contacting walls, (d) the incorporation into pre-existing vessels of the interstitial structures surrounded by patent vessel loops formed by sprouting angiogenesis from these pre-existing vessels, (e) thrombus fragments or microthrombi originating transitional cores covered by reoriented ECs from the vessel wall, and (f) combinations of some of these mechanisms [26,39,41,45,46,200,201,202].

In the initially described and better-known mechanism, pericytes incorporate after changes in ECs of the pre-existing vessels [22]. Thus, after the establishment of interendothelial contacts between opposite vessel walls and formation of intravascular endothelial bridges (endothelial cell intussusception, nascent pillars) (stage 1), reorganization of EC junctions, EC bilayer arrangement, and formation of the central virtual core (pillar perforation) (stage II) [203], collagen and pericytes integrate into this core (stage III) (Figure 11A–C). The pressure exerted by pericytes can also facilitate interendothelial contacts from the opposite vessel walls [39]. In this mechanism, the basement membrane of the parent vessel is conserved. In a variant of this mechanism, denominated “inverse sprouting”, the recruitment of pericytes into the pillar core occurs after (a) focal degradation of the basement membrane restricted to the point at which the endothelial ridge originates, (b) attachment of ECs to the perivascular collagen fascicles at the point of basement membrane degradation, and (c) retraction of the attached ECs, with participation of the actin cytoskeleton, and incorporation of collagen into the core of nascent pillars [36]. A similar incorporation of pericytes occurs in the virtual space formed by merged adjacent capillaries. Likewise, pericytes are integrated in the transitional core formed by thrombus components after the thrombus has been lined by reoriented ECs from the vessel wall.

In pillar formation mechanisms through the splitting of pre-existing pillars or vessel loop development, pericytes may already be present in the interstitial tissue structure that subsequently forms the pillar core when either the refolded endothelium in the pre-existing pillar or the surrounding vessel loops becomes patent.

Numerous peg-and-socket junctions are frequently established between pericytes and ECs in pillars (Figure 11D). In these junctions, pericytes form the peg and ECs form the socket in their abluminal surface [3,74,204,205]. A similar increase in this type of union has been described between vascular smooth muscle cells and ECs in some processes, such as arterial intimal thickening [206].

The behavior of pericytes during angiogenesis contributes to the heterogeneity of angiogenesis and blood vessel maturation in human tumors [207,208]. In addition, pericyte participation can be very irregular in tumors in which intussusceptive angiogenesis participates in association with sprouting angiogenesis. For example, the disproportion in pericyte/EC proliferation during intussusceptive angiogenesis participates in the formation of bizarre vessels (vascular clusters, vascular garlands, and glomeruloid bodies) in glioblastoma [149].

## 8. CD34+SCs in Intussusceptive Angiogenesis

CD34+SCs/TCs show few changes in interstitial tissue during intussusceptive angiogenesis since the main events in this type of angiogenesis occur in the lumen of pre-existing vessels, with preservation of vascular blood flow and no increased vascular permeability, EC invasion of the interstitium, or interstitial tissue degradation. SCs have been described in the core of pillars in advanced stages of evolution [39], although CD34+SCs/TCs are rarely observed in the pillar core (Figure 11F). However, we have observed SCs, or their processes, in the core of vascular wall folds during intussusceptive angiogenesis (Figure 11E). Therefore, perivascular CD34+SCs/TCs may invaginate, together with pericytes, in the core of the pillars, giving rise to these SCs, a possibility that requires further study.

## 9. General Considerations about the Behavior of Pericytes and CD34+SCs/TCs in Sprouting and Intussusceptive Angiogenesis

Differentiation of pericyte and CD34+SC/TC findings according to angiogenesis type may be difficult since sprouting and intussusceptive angiogenesis can be complementary mechanisms, with synergistic interactions [25,27,194,209,210,211]. Likewise, intussusceptive angiogenesis can participate in capillary expansion and vessel remodeling following sprouting angiogenesis [27,209].

Pericytes and CD34+SCs/TCs are, to a greater or lesser extent, involved in angiogenesis. The main determinant of their behavior in sprouting and intussusceptive angiogenesis is the growth path of ECs: (a) growing toward the interstitium with interstitial tissue morphogenic findings in sprouting angiogenesis and (b) extending toward the lumen of the vessel itself with intraluminal morphogenic findings in intussusceptive angiogenesis. An important fact that emerges from this different form of growth is that the lumen and blood flow must be at least partially preserved for intussusceptive angiogenesis to take place whereas sprouting angiogenesis can occur with or without blood flow preservation in parent vessels. All this entails the non-interruption of functionality during the formation of neovessels in intussusceptive angiogenesis, while a certain time is needed for the vascular buds to integrate into the vascular system in sprouting angiogenesis.

In sprouting and intussusceptive angiogenesis, pericytes, perivascular CD34+SCs/TCs, and homing cells from the bone marrow (MSCs and monocytes/fibrocytes) form a niche and transient point of precursor cells. Interactions between pericytes and ECs through autocrine and paracrine pathways act on the behavior of these cells during sprouting and intussusceptive angiogenesis. Likewise, together with CD34+SCs/TCs and transmigrating cells from the bone marrow, they form a common substrate with multiple interactions (cell–cell contacts and soluble factors). These pathways control the quiescent and angiogenic stages of the microvasculature and regulate cell mobilization, proliferation, recruitment, and differentiation, as well as vessel destabilization and stabilization [212,213].

The main events of pericytes and CD34+SCs/TCs in sprouting and intussusceptive angiogenesis are summarized in Figure 12. The following occurs in sprouting angiogenesis: (a) detachment of pericytes and CD34+SCs/TCs from the vessel wall, (b) pericyte mobilization and proliferation, (c) pericyte recruitment in the endothelial sprouts, with basal membrane deposition and vessel stabilization, (d) proliferation of CD34+SCs/TCs, some of which acquire lipofibroblast-like characteristics, and both can co-express CD34 and αSMA, behaving as a source of myofibroblasts, and (e) increased number of stromal cells around vessels in regression with platelet aggregates. In intussusceptive angiogenesis, pericytes are incorporated into the pillar cores. The intussusceptive mechanism dictates the incorporation of pericytes. For example, they extend into the virtual cores of previously formed endothelial bridges or form part of intraluminal pillar cores when vessel loops, originating from two points of pre-existing vessels and surrounding them, become patent. Numerous peg-and-socket junctions are established between pericytes and endothelial cells. Although CD34+SCs/TCs can be incorporated into the pillar cores, it occurs less frequently. They also present few changes around the vessels and in the interstitium.

Therefore, the proliferation of ECs, pericytes, and CD34+SCs/TCs is usually much higher in sprouting angiogenesis than in intussusceptive angiogenesis. The scarce proliferation of these cells in intussusceptive angiogenesis is compensated for by the spread and thinning of ECs and the incorporation of pericytes or their processes into the intraluminal pillars by migration or extension, respectively [25,34,35,42,214]. Thus, sprouting angiogenesis is largely involved in tissue repair through granulation tissue development, as well as in tumor stroma formation. During these processes, proliferating CD34+SCs/TCs lose CD34 expression and gain αSMA expression, differentiating into myofibroblasts (cancer-associated fibroblasts, CAF, in tumors) [150,151,215]. Conversely, the participation of CD34+SCs/TCs in the intussusceptive process is rare. All this entails that the duration time and the metabolic cost in sprouting angiogenesis are higher than in intussusceptive angiogenesis with little proliferation and without invasive behavior toward the interstitium [25,34,35]. However, EC and pericyte proliferation can occur when the divided vessels expand their lumen. Likewise, in some tumors, such as glioblastoma, an intense proliferation of pericytes can be observed in aberrant vessels with intussusceptive angiogenesis [149].

We have highlighted the best-known and general differences in the behavior of pericytes and CD34+SCs/TCs depending on whether angiogenesis is through sprouting or intussusception. In this comparison, other aspects are more difficult to establish. For example, the difference between pericytes and CD34+SCs/TCs according to angiogenesis during aging has been better investigated in conventional sprouting angiogenesis than in intussusceptive angiogenesis. Thus, there are several studies on the aging-related modifications of angiogenesis [216,217,218], and it has been hypothesized that pericytes in aged networks have an increased stabilization phenotype and decreased proangiogenic function [216]. Likewise, VEGF+ telocytes (CD34+SCs/TCs) were seen in the stroma of the prostate, possibly contributing to angiogenesis, during aging-related changes [58]. Other important issues are the behavior and functions of these cells during tumor formation and in healthy and disease organs. Some of these aspects have been extensively reviewed, although with few comparisons depending on the type of angiogenesis studied here. In addition to neo-vessel formation in both types of angiogenesis, there is greater participation of these perivascular cells in the tumor stroma formation in sprouting angiogenesis as mentioned above, while they have an important role in vessel arborization, branching remodeling, pruning, and compartmentalization, as well as in the formation of intravascular septa in intussusceptive angiogenesis [25,38,42,43,44,45,219]. Therefore, the resulting structures by these procedures can be very evident during development, in diseases of different organs with vessel involvement, and in the morphogenesis of vessel tumors and pseudotumors [73,200,215].

Tumor recurrence after antiangiogenic or antineoplastic treatment can occur by a transient switch from sprouting to intussusceptive angiogenesis [195]. Therefore, a better understanding of the behavior, function, and modulation of pericytes, CD34+SCs/TCs, and derived cells, genes, and signals pathways involved in these angiogenic processes, as well as of those that regulate the transition from sprouting to intussusceptive angiogenesis, is of interest for the development of new antiangiogenic therapies and to prevent tumor recurrences [220,221,222].

## 10. Conclusions

Throughout this work we have reviewed the behavior of pericytes and CD3+SCs/TCs in sprouting and intussusceptive angiogenesis, comparing the main findings of both types of cells in these angiogenic processes. The differences mainly depend on the path followed by the endothelial and perivascular cells and the structures formed by them: toward the interstitium with formation of vascular buds and neovessels in sprouting angiogenesis and toward the vessel lumen with formation of intravascular pillars and vessel division in intussusceptive angiogenesis. Thus, the main differences in the behavior of pericytes and CD34+SCs/TCs in both types of angiogenesis are: (a) detachment of the vessel wall and migration in the interstitial tissue occurs in sprouting angiogenesis, whereas there is extension or incorporation (predominantly of pericytes and their processes) into the intravascular pillar core in intussusceptive angiogenesis; (b) perivascular cell proliferation and recruitment are more intense and with a greater energy cost in sprouting angiogenesis than in intussusceptive angiogenesis; (c) perivascular CD34+SCs/TCs behave as precursor cells in repair and tumor stroma formation during sprouting angiogenesis, whereas they are little involved in these processes during intussusceptive angiogenesis; (d) these mechanisms are regulated by cell–cell contacts, numerous signaling pathways, and factors such as hypoxia and blood flow, which require future studies on their presentation and balance for a better understanding of the evolution of angiogenesis from pre-existing vasculature into one type or the other.

## Figures and Tables

**Figure 1 ijms-23-09010-f001:**
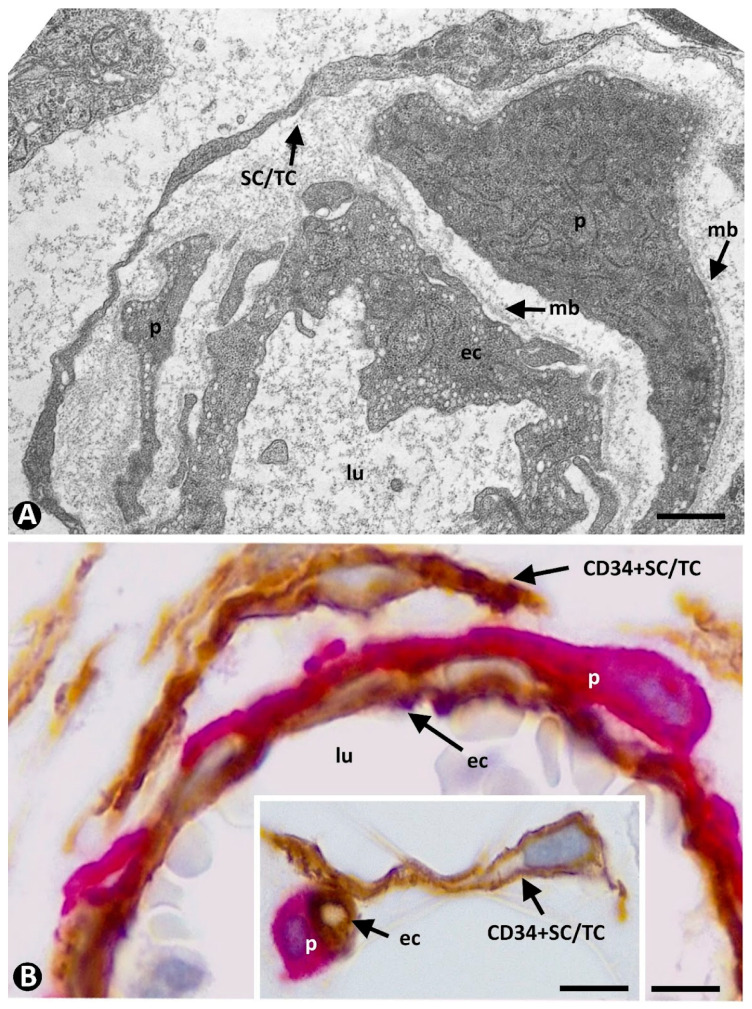
(**A**): Ultrastructural image of a small vessel of the pericytic microvasculature with endothelial cells (ec) in the intima, pericytes (p) in the media, and processes of stromal cells/TCs (SC/TC) in the adventitia. Note a basal membrane (mb) shared by endothelial cells and pericytes. (**B**) and insert: Endothelial cells (ec, brown) in the intimal layer, pericytes (p, red) in the media layer, and CD34+SCs/TCs (brown) in the adventitia in a venule (**B**) and a capillary (insert of (**B**)) in a very initial stage of angiogenesis during granulation tissue formation. Vessel lumen: lu. (**A**): Ultrathin section. Uranyl acetate and lead citrate. (**B**) and insert: Double immunochemistry for CD34 (brown) and αSMA (red). Bar: (**A**): 0.8 µm; (**B**): 8 µm.

**Figure 2 ijms-23-09010-f002:**
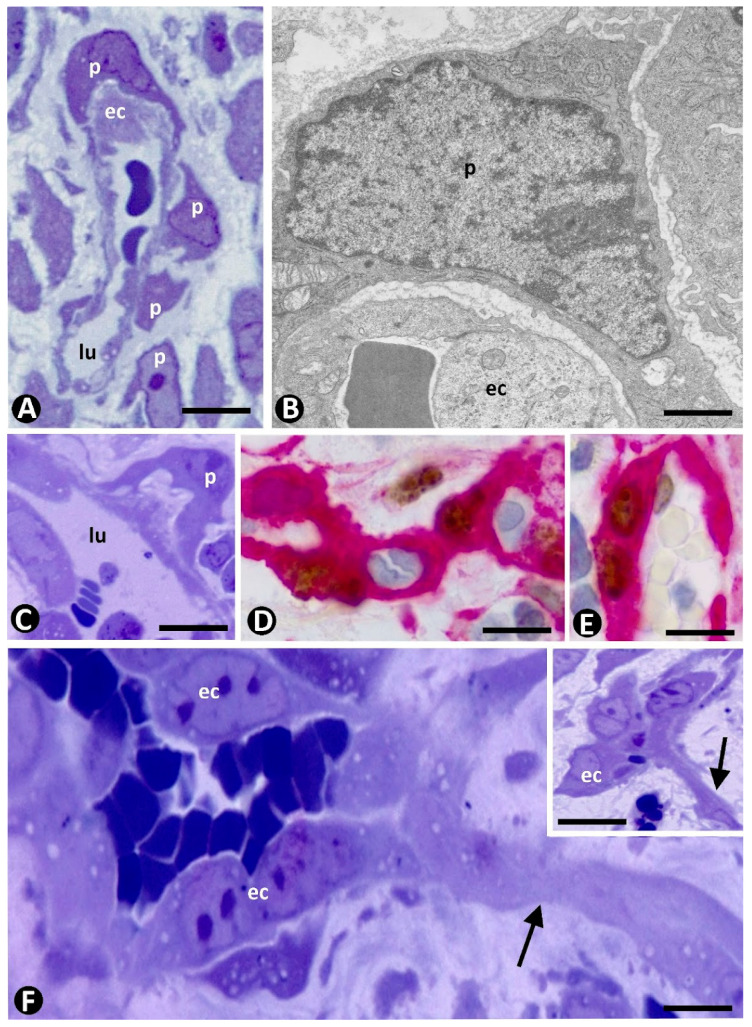
Pericyte during the initial stage of sprouting angiogenesis. (**A**–**C**): Presence in the parent vessels of pericytes (p) increased in size, with shortening of their processes, voluminous nuclei, prominent nucleoli, and abundant intracytoplasmic ribosomes. (**D**,**E**): Nuclear expression of ki-67 (brown) in pericytes with cytoplasmic expression of αSMA (red). (**F**) and insert: Sprouts of endothelial cells (arrows) devoid of pericytes. Endothelial cell: ec. Vessel lumen: lu. (**A**,**C**,**F**) and insert of (**F**): Semithin sections. Toluidine blue staining. (**B**): Ultrathin section. Uranyl acetate and lead citrate. (**D**,**E**): Double immunochemistry for anti-ki-67 and αSMA. Bar: (**A**): 12 µm; (**B**): 0.8 µm; (**C**–**E**): 15 mm; (**F**): 10 µm.

**Figure 3 ijms-23-09010-f003:**
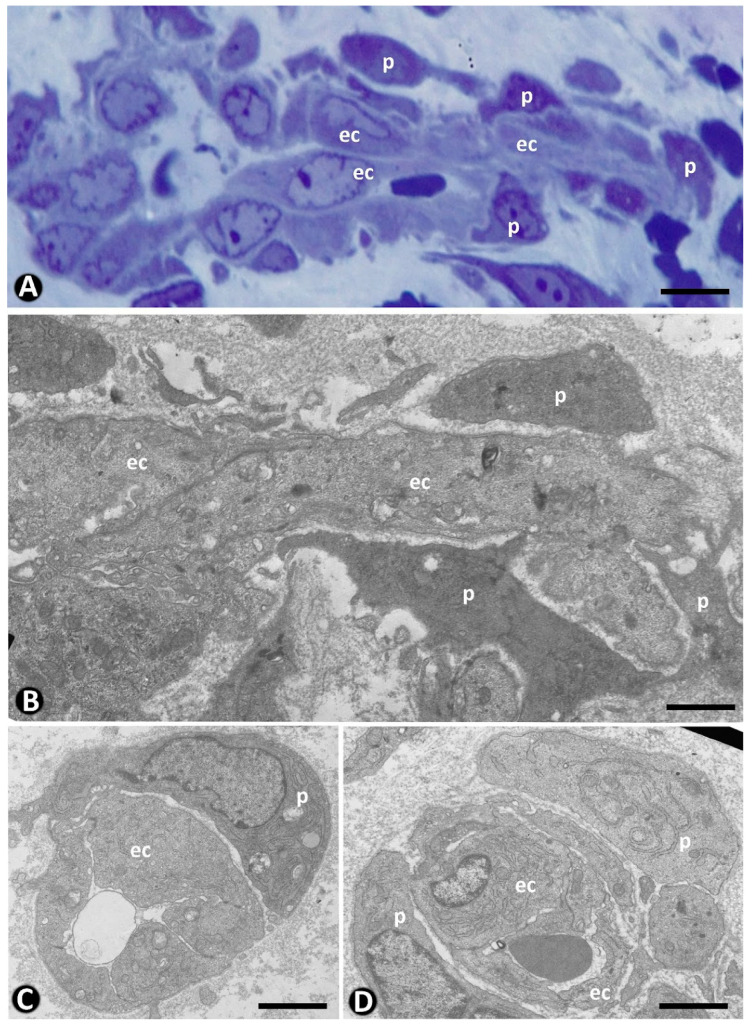
Pericyte recruitment in the vascular sprouts. Pericytes (p) observed around vascular sprouts in semithin (**A**) and ultrathin (**B**–**D**) sections. In (**B**–**D**), the ultrastructural images of the vascular sprouts correspond to longitudinal (**B**) and transverse (**C**,**D**) sections. Endothelial cell: ec. A: Semithin section. Toluidine blue staining. (**B**–**D**): Ultrathin sections. Uranyl acetate and lead citrate. Bar: (**A**): 10 µm; (**B**): 0.8 µm; (**C**,**D**): 1.5 µm.

**Figure 4 ijms-23-09010-f004:**
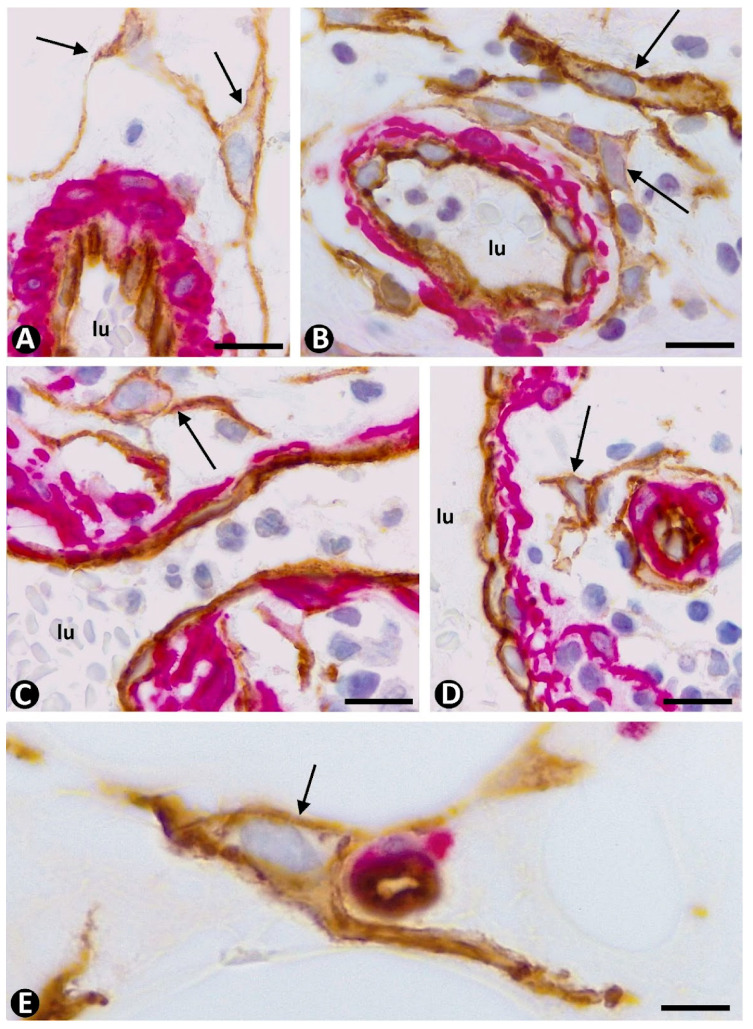
(**A**–**D**): In previous stages of sprouting angiogenesis, CD34+SCs/TCs (brown, arrows) are seen in edematous spaces around different sized parent vessels, which show ECs (brown) and mural cells (red). Vessel lumen: lu. CD34+SCs/TCs tend to separate from the vessel wall. (**E**): A hypertrophied CD34+/SC/TC observed near a capillary with a small lumen. Double immunochemistry for CD34 (brown) and αSMA (red). Hematoxylin counterstain. Bar: (**A**–**D**): 30 µm; (**E**): 8 µm.

**Figure 5 ijms-23-09010-f005:**
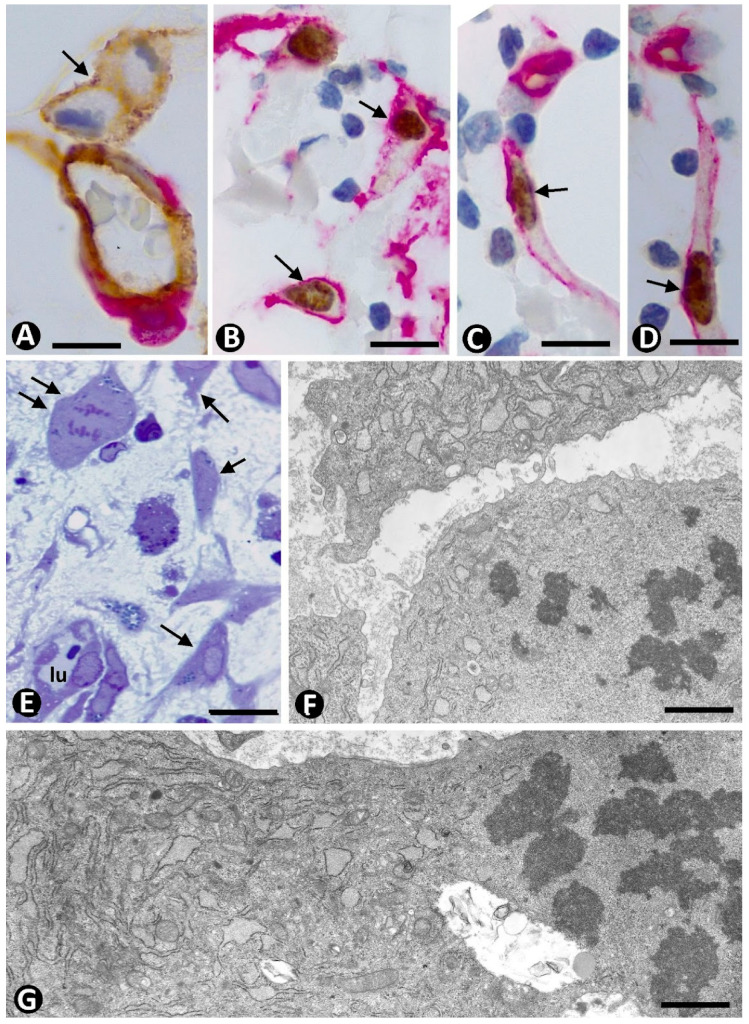
Mitosis and high proliferative index in stromal cells during sprouting angiogenesis. (**A**): A mitosis in telophase observed in a CD34+SC/TC around a vessel, in which endothelial cells (brown) and pericytes (red) are seen. (**B**–**D**): CD34+SCs/TCs (red, with peripheral expression of CD34) expressing anti-ki-67 (brown, with nuclear expression). (**E**): Hypertrophied stromal cells (arrows), one in mitosis (double arrow), around a vessel. (**F**,**G**): Ultrastructural characteristics of stromal cells in mitosis. Note the presence of chromosomes and abundant endoplasmic reticulum. Vessel lumen: lu. (**A**): Double immunochemistry staining for CD34 (brown) and αSMA (red). Hematoxylin counterstain. (**B**–**D**): Double immunochemistry staining for CD34 (red) and ki-67. Hematoxylin counterstain. (**E**): Semithin section. Toluidine blue staining. (**F**,**G**): Ultrathin sections. Uranyl acetate and lead citrate. Bar: (**A**–**D**): 30 µm; (**E**): 12 µm; (**F**): 1 µm; (**G**): 0.8 µm.

**Figure 6 ijms-23-09010-f006:**
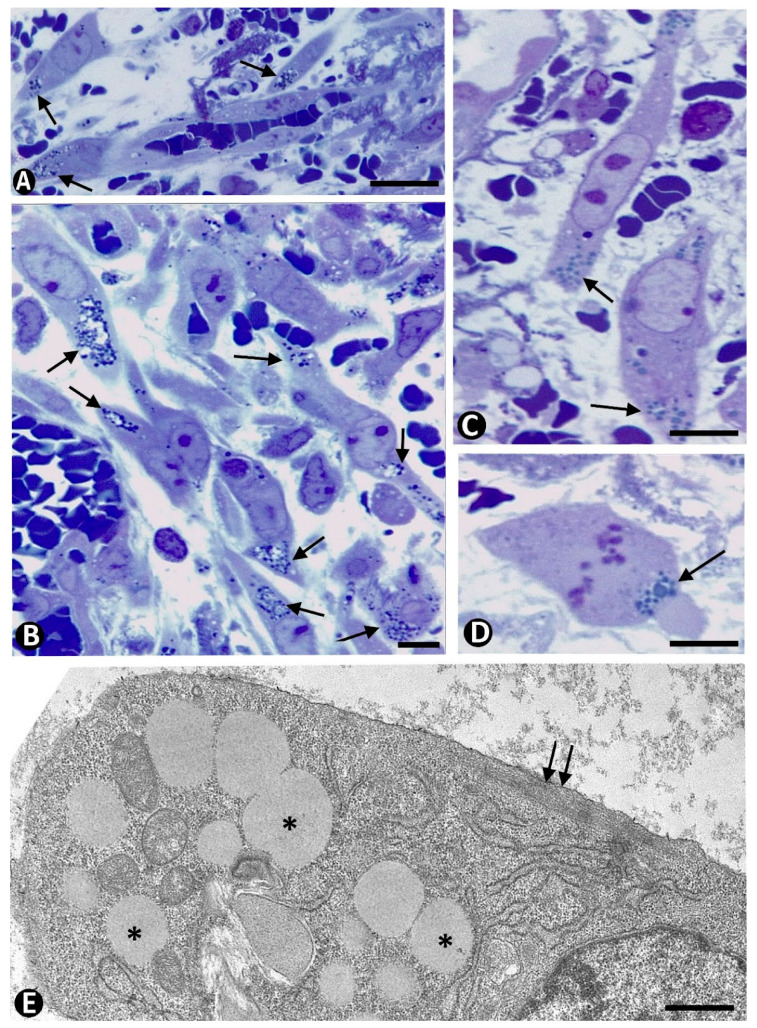
Presence of intracytoplasmic lipid droplets in some stromal cells (lipid-laden cells, lipid-interstitial cells, lipofibroblast-like cells). Note groups of lipid droplets in semithin sections (**A**–**D**, arrows). One cell in mitosis (**D**) and the ultrastructural characteristics of a lipid-interstitial cell with lipid droplets (asterisks) and a small peripheral band of filaments with dense bands (double arrow). (**A**–**D**): Semithin sections. Toluidine blue staining. (**E**): Ultrathin section. Uranyl acetate and lead citrate. Bar: (**A**): 45 µm; (**B**): 12 µm; (**C**,**D**): 15 µm; (**E**): 0.8 µm.

**Figure 7 ijms-23-09010-f007:**
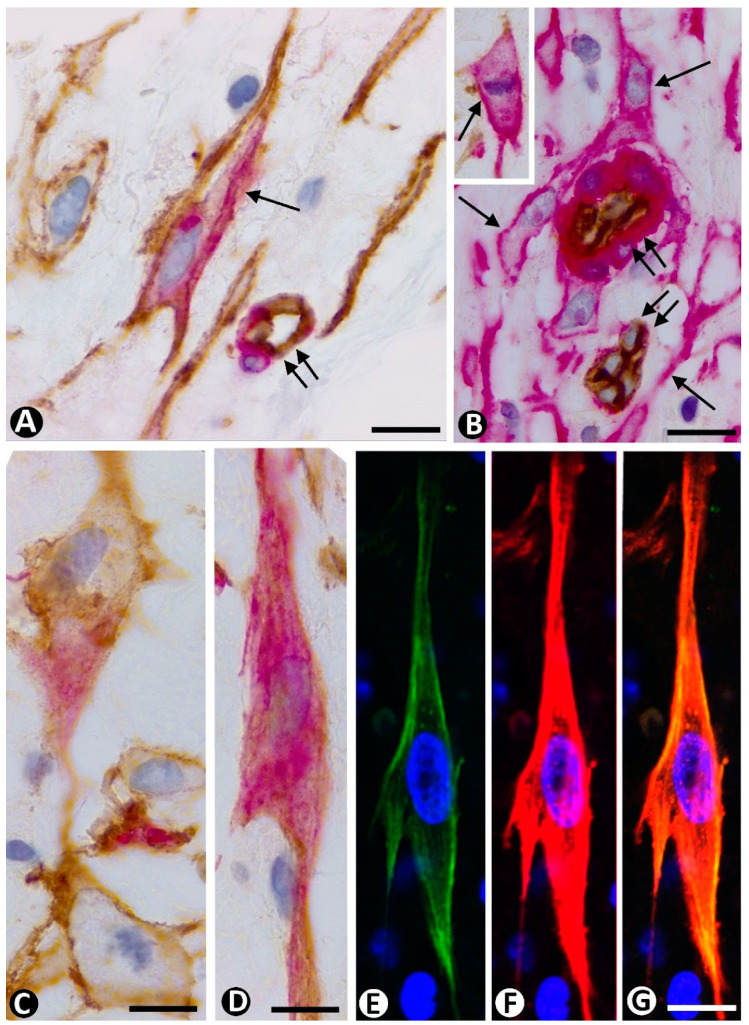
(**A**): Presence of CD34+SCs/TCs (brown) and a stromal cell (arrow) suggesting expression of CD34 (brown) and αSMA (red) around a small vessel (double arrow) with an endothelial cell (brown) and a pericyte (red). (**B**): All stromal cells (arrows) around vessels (double arrow) are myofibroblasts expressing αSMA (red). Insert of (**B**): A myofibroblast in mitosis (arrow). (**C**,**D**): Details of stromal cells suggesting transitional cell forms between CD34+SCs/TCs and myofibroblasts by double immunochemistry for CD34 and αSMA. (**E**–**G**): Co-expression of CD34 and αSMA observed in a stromal cell by double immunofluorescence labeling for CD34 and αSMA. (**A**–**D**): Double immunochemistry for CD34 (brown) and αSMA (red). (**E**–**G**): Double immunofluorescence in confocal microscopy for CD34 (**E**, green), αSMA (**F**, red), and merged (**G**). Bar: (**A**): 8 µm; (**B**): 30 µm; (**C**–**G**): 8 µm.

**Figure 8 ijms-23-09010-f008:**
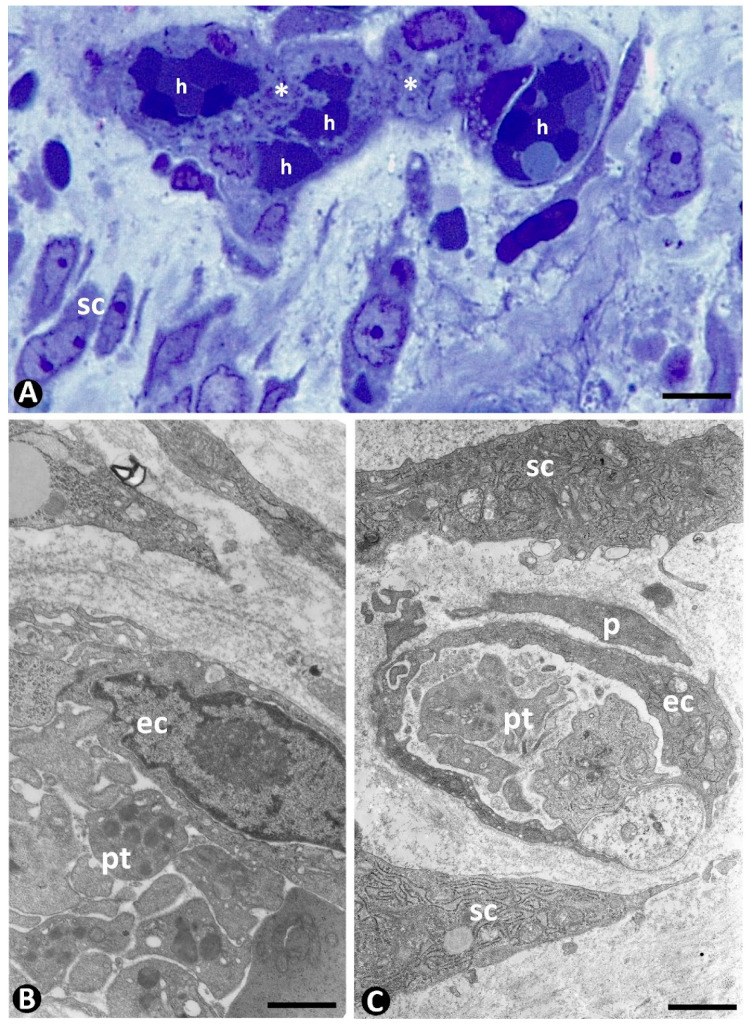
Intravascular accumulation of platelets during vessel regression. Platelet aggregates (asterisks) and red blood cells (h) in regressing vessels and interstitial/stromal cells in a semithin section. (**B**,**C**): Ultrastructural images of intravascular platelets (pt). Endothelial cells: ec. Pericyte: p. Stromal cells: sc. (**A**): Semithin section. Toluidine blue staining. (**B**,**C**): Ultrathin sections. Uranyl acetate and lead citrate. Bar: (**A**): 10 µm; (**B**,**C**): 1 µm.

**Figure 9 ijms-23-09010-f009:**
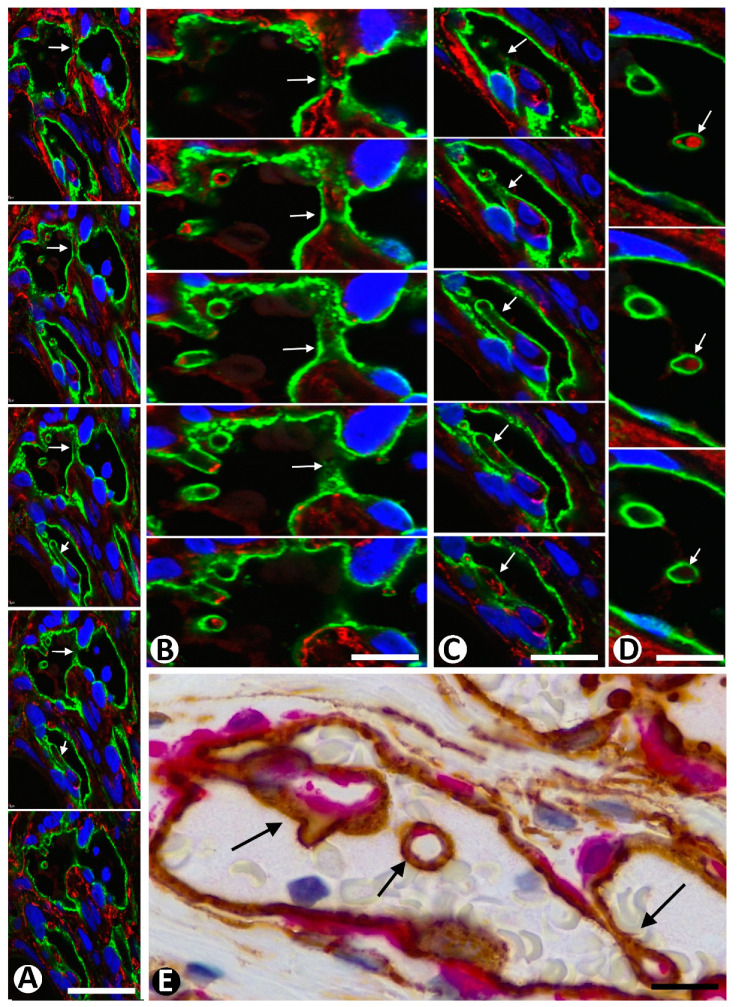
(**A**–**C**): Several intravascular pillars (arrows), which appear and disappear in sequential views, are shown in confocal microscopy in A, and details of some in (**B**,**C**). (**D**): Sequential views of cross-sectioned pillars in confocal microscopy (arrows). Note the cover of pillars formed by endothelial cells (green) and the presence in the core of some pillars of collagen IV (red). (**E**): Intravascular pillars (arrows), observed by immunochemistry, showing endothelial cells (brown) forming the pillar cover and pericytes (red) in the pillar core. (**A**,**D**): Double immunofluorescence labeling for CD34 (green) and anti-collagen IV (red) in sequential views in confocal microscopy. (**E**): Double immunochemistry for CD34 (brown) and αSMA (red). Bar: (**A**–**D**): 10 µm; (**E**): 15 µm.

**Figure 10 ijms-23-09010-f010:**
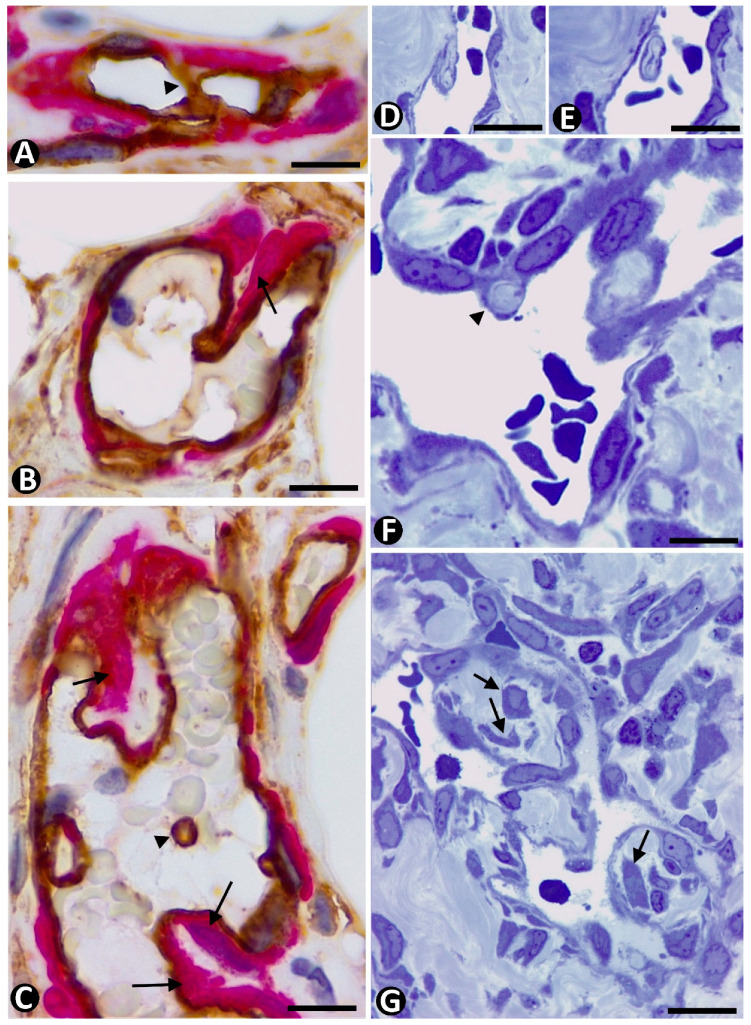
Incorporation of pericytes into intravascular pillars (stage III) after development of interendothelial contacts and bridges between opposite vessel walls (stage I) and pillar core formation (stage II). (**A**–**C**): An interendothelial bridge between opposite walls of a vessel (**A**, arrowhead), pericytes (red) incorporating in the pillar core (**B**,**C**, arrows), and pillars whose formation begins without pericyte incorporation (**C**, arrowheads) observed by immunochemistry. (**D**–**G**): Observation in semithin sections of the incorporation of pericytes in pillars. Note some pillars whose formation begins without pericytes, which are yet to be incorporated (**D**–**F**, arrowhead), and other pillars in which pericytes or their processes are present in the pillar core (**F**,**G**, arrows). (**A**–**C**): Double immunochemistry for CD34 (brown) and αSMA (red). (**E**–**G**): Semithin sections. Toluidine blue staining (**D**,**E** correspond to a pillar sectioned at two heights). Bar: (**A**,**B**): 30 µm; (**C**,**F**): 15 µm; (**D**,**E**): 10 µm; (**F**): 15 µm; (**G**): 45 µm.

**Figure 11 ijms-23-09010-f011:**
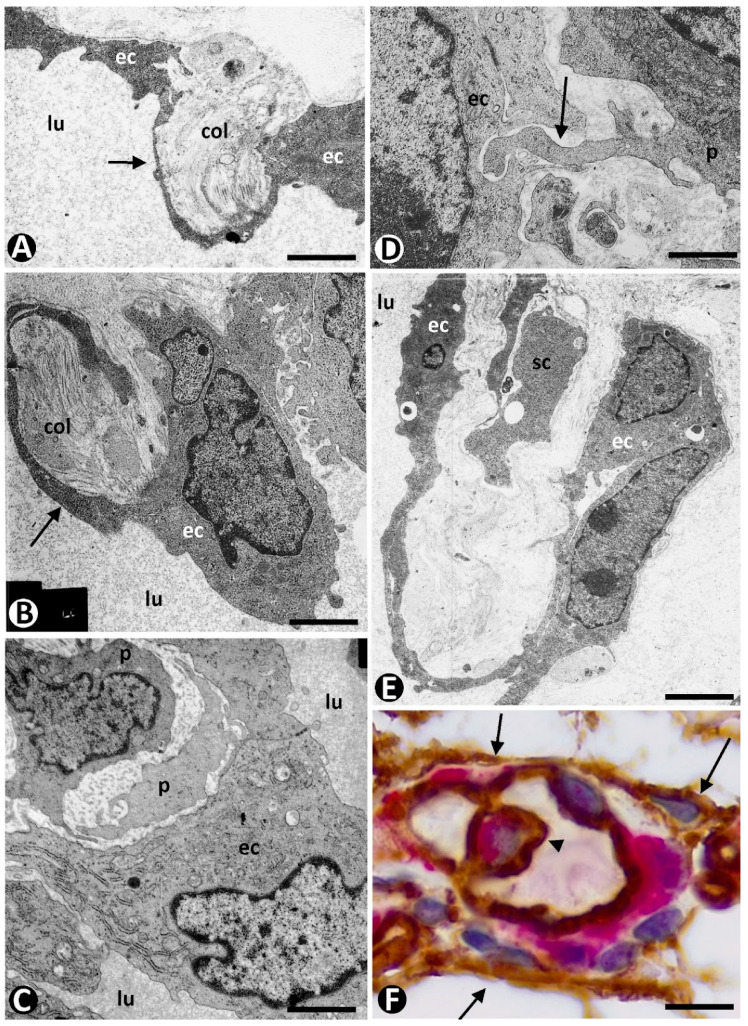
Pericytes and stromal cells in intussusceptive angiogenesis. (**A**–**C**): Intravascular pillars in successive stages of formation. Note in (**A**,**B**), collagen (col) incorporating in the core of pillars (arrows) surrounded by a cover formed by extending endothelial cells. In (**C**), pericytes (p) are present in the core of a pillar surrounded by prominent endothelial cells (ec). (**D**): A peg-and-socket junction (arrow) observed between a pericyte (p) and an endothelial cell (ec). (**E**): A pillar surrounded by endothelial cells (ec) and in the core of collagen and some stromal cell (sc) processes. (**F**): Immunohistochemistry demonstration of CD34+/SCs (brown) (arrows) around a vessel wall with a pillar (arrowhead) made up of a cover formed by ECs and a core with pericytes (red). lu: vessel lumen. (**A**–**E**): Ultrathin sections, Uranyl acetate, and lead citrate. (**F**): Double immunohistochemistry for CD34 (brown) and αSMA (red). Bar: (**A**,**B**): 2.5 µm; (**C**): 1 µm; (**D**): 2 µm; (**E**): 2.5 µm; (**F**): 15 µm.

**Figure 12 ijms-23-09010-f012:**
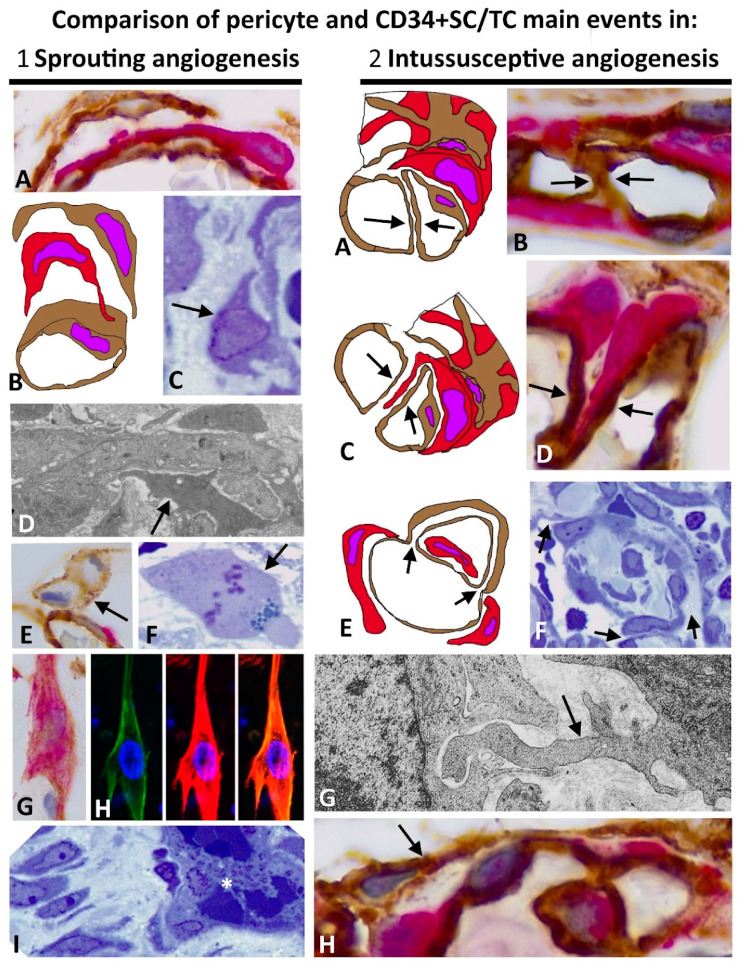
Schematic and microphotographic representation of the main aspects of the behavior of pericytes and CD34+SCs/TCs in sprouting (1) and intussusceptive (2) angiogenesis. (1) In sprouting angiogenesis, pericytes and CD34+SCs/TCs of the activated pre-existing vessels (**A**) detach from the vessel wall (**B**,**C**, arrow). Pericytes proliferate and are recruited in the endothelial sprouts (**D**, arrow), leading to EC survival, basal membrane deposition, and vessel stabilization. CD34+SCs/TCs proliferate (**E**, arrow), some acquire lipofibroblast-like characteristics (**F**, arrow) and show coexpression of CD34 and αSMA (**G**,**H**), transforming into αSMA+ myofibroblasts. Many newly formed vessels regress, showing intraluminal aggregate of platelets (asterisk) with increased interstitial myofibroblasts (**I**). (2) In intussusceptive angiogenesis, pericytes, and less frequently CD34+SCs/TCs may form part of the intravascular pillar cores. Incorporation depends on the mechanism of pillar development, for example, by extension after endothelial bridge formation (**A**–**D**, arrows) or by engulfment through vessel loop formation (**E**,**F**, arrows). Numerous peg-and-socket junctions are established between pericytes and ECs (**G**, arrow). Generally, perivascular and interstitial CD34+SCs/TCs remain with scarce modifications (**H**, arrow). Bar: sprouting (**A**,**G**,**H**): 8 µm; (**C**): 12 µm; (**D**): 0.8 mm; (**E**): 30 µm; (**F**): 15 µm; (**I**): 10 µm. intussusceptive (2): (**B**,**D**): 30 µm; (**F**): 45 µm; (**G**): 2 µm; (**H**): 15 µm.

## Data Availability

All the data are reported in the present paper.

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
