# Peer review of "Comparison of the Behavior of Perivascular Cells (Pericytes and CD34+ Stromal Cell/Telocytes) in Sprouting and Intussusceptive Angiogenesis"

_ijms, 2022, doi:10.3390/ijms23169010_

Round 1
Reviewer 1 Report
Díaz-Flores et al have neatly summarized the behaviour of pericytes and CD3+SCs/TCs in sprouting and intussusceptive angiogenesis and managed to draw a parallel between both types of cells in these angiogenic processes.
The manuscript is well written, clear, and easy to understand. There is indeed little literature on the overall subject, and the authors managed to highlight aspects that merit further consideration. The study is well-done and the images are clear, depicting data thoroughly explained within the text. I found no issues related to language, grammar, and typing mistakes.
In the Abstract, the authors state that “the dysfunction of these mechanisms is involved in several diseases, including neoplasms, with therapeutic implications”. Could they perhaps mention one potential therapeutic strategy involving sprouting and intussusceptive angiogenesis of perivascular cells?
Overall, I am pleased with this article and, following minor revision, I believe it can be published in IJMS.
Author Response
We have added the following sentence in the general considerations about the behavior of pericytes and CD34+SCs/TCs in sprouting and intussusceptive angiogenesis to answer one potential therapeutic strategy involving these processes:
Tumor recurrence after antiangiogenic or antineoplastic treatment can occur by a transient switch from sprouting to intussusceptive angiogenesis [195]. Therefore, a better understanding of the behavior, function, and modulation of pericytes, CD34+SCs/TCs, and derived cells, genes, and signals pathways involved in these angiogenic processes, as well as of those that regulate the transition from sprouting to intussusceptive angiogenesis, is of interest for the development of new antiangiogenic therapies and to prevent tumor recurrences [220-222].
Reviewer 2 Report
It's a comprehensive review that compares the behavior of pericytes and cd34+ stromal cell/telocytes as perivascular cells in sprouting and intussusceptive angiogenesis.
Would you please answer these questions?
1.What is the difference between these cells according to angiogenesis during aging?
2. What is the difference between these cells during tumor formation?
3. What is the difference between these cells' behavior and functions in healthy and diseased organs?
Author Response
We have added the following sentences in the general considerations about the behavior of pericytes and CD34+SCs/TCs in sprouting and intussusceptive angiogenesis to answer the three questions raised:
We have highlighted the best-known and general differences in the behavior of pericytes and CD34+SCs/TCs depending on whether angiogenesis is through sprouting or intussusception. In this comparison, other aspects are more difficult to establish. For example, the difference between pericytes and CD34+SCs/TCs according to angiogenesis during aging has been better investigated in conventional sprouting angiogenesis than in intussusceptive angiogenesis. Thus, there are several studies on the aging-related modifications of angiogenesis [216-218], and it has been hypothesized that pericytes in aged networks have an increased stabilization phenotype and decreased proangiogenic function [216]. Likewise, VEGF+ telocytes (CD34+SCs/TCs) were seen in the stroma of the prostate, possibly contributing to angiogenesis, during aging-related changes [58]. Other important issues are the behavior and functions of these cells during tumor formation and in healthy and disease organs. Some of these aspects have been extensively reviewed, although with few comparisons depending on the type of angiogenesis studied here. In addition to neo-vessel formation in both types of angiogenesis, there is greater participation of these perivascular cells in the tumor stroma formation in sprouting angiogenesis as mentioned above, while they have an important role in vessel arborization, branching remodeling, pruning, and compartmentalization, as well as in the formation of intravascular septa in intussusceptive angiogenesis [25, 38,42-45, 219]. Therefore, the resulting structures by these procedures can be very evident during the development, in diseases of different organs with vessel involvement, and in the morphogenesis of vessel tumors and pseudotumors [73,200,215].